# No Impact of PolySia-NCAM Expression on Treatment Response in Neuroendocrine Neoplasms of the Lung

**DOI:** 10.3390/cancers14184376

**Published:** 2022-09-08

**Authors:** Daniel Gagiannis, Anna Scheil, Sarah Gagiannis, Carsten Hackenbroch, Ruediger Horstkorte, Konrad Steinestel

**Affiliations:** 1Department of Pulmonology, Bundeswehrkrankenhaus Ulm, 89081 Ulm, Germany; 2Department of Neurology, Bundeswehrkrankenhaus Ulm, 89081 Ulm, Germany; 3Department of Radiology, Bundeswehrkrankenhaus Ulm, 89081 Ulm, Germany; 4Institute for Physiological Chemistry, Medical Faculty, Martin-Luther-University Halle-Wittenberg, 06114 Halle (Saale), Germany; 5Institute of Pathology and Molecular Pathology, Bundeswehrkrankenhaus Ulm, 89081 Ulm, Germany

**Keywords:** polySia-NCAM, neuroendocrine lung tumors, carcinoid, SCLC, LCNEC

## Abstract

**Simple Summary:**

Polysialic acids (polySia) are localized on the neuronal cell adhesion molecule (NCAM). They are expressed on numerous tumors of neural crest origin. These include lung neuroendocrine tumors such as atypical carcinoid, large cell neuroendocrine and small cell carcinomas. Interfering with polySia is considered a potential approach in the development of tumor therapies. In this study, we investigated whether polySia expression has an impact on disease progression, treatment response, and prognosis. To this end, tissue samples from 28 patients were analyzed by immunohistochemistry for polySia-NCAM presence. In conclusion, NCAM-polySia is not very useful as a prognostic factor for poor disease outcome. However, it is still interesting as a therarpeutic target for individual tumor therapy, as a majority of patients (78.6%) showed a strong staining signal for NCAM-polySia.

**Abstract:**

Background: Polysialic acids (abbr. polySia) are found on numerous tumors, including neuroendocrine lung tumors. They have previously been shown to impact metastatic potential, as they can influence the signaling and adhesion properties of neuronal cell adhesion molecules (abbr. NCAM) and other cell adhesion molecules. Therefore, the aim of this small pilot study was to analyze whether there was a correlation between polySia-NCAM expression and specific clinical or histopathologic characteristics, and if polySia-NCAM expression had an impact on treatment response, disease progression and prognosis of lung neuroendocrine neoplasms. Methods: This work was based on an analysis of 28 digitized patient records and corresponding patient samples. The response to therapy was radiologically determined at the time of diagnosis and at certain intervals during therapy following the current RECIST1.1 and volumetric sphere calculation. To analyze whether polySia-NCAM expression had prognostic relevance, polySia-NCAM-positive and -negative cases were compared in a Kaplan-Meier survival analysis. Findings: A majority of 78.6% lung neuroendocrine neoplasms showed a strong staining signal for polySia-NCAM. There was a significant correlation between expression and histopathological grade (*p* = 0.0140), since carcinoids were less likely polySia-NCAM-positive compared to small cell lung carcinoma (abbr. SCLC) and large cell neuroendocrine carcinomas of the lung (abbr. LCNEC). There was no significant association between polySia-NCAM expression and clinical characteristics (age: *p* = 0.3405; gender: *p* = 0.6730; smoking history: *p* = 0.1145; ECOG: *p* = 0.1756, UICC8 stage: *p* = 0.1182) or radiologically determined disease progression, regardless of the criteria used to categorize response (RECIST 1.1: *p* = 0.0759; sphere: *p* = 0.0580). Furthermore, polySia-NCAM expression did not affect progression-free survival (*p* = 0.4198) or overall survival (*p* = 0.6918). Interpretation: PolySia-NCAM expression was more common in high-grade compared to low-grade neuroendocrine neoplasms of the lung; however, this small pilot study failed to show an association between polySia-NCAM expression and response to therapy.

## 1. Key Message

-What is the key question?

Is there a correlation between polySia-NCAM expression and clinic-pathological characteristics of lung neuroendocrine neoplasms, and does polySia-NCAM expression have an impact on treatment response, disease progression and prognosis of the disease?

-What is the bottom line?

Although high-grade neuroendocrine neoplasms were more likely to express polySia-NCAM compared to carcinoids, our small pilot study showed no significant correlation between polySia-NCAM expression and clinical characteristics or radiologically determined disease progression. Accordingly, polySia-NCAM was not a prognostic factor for poor disease outcome.

-Why read on?

Due to the established role of polySia-NCAM in tumor progression and metastasis, it was assumed that this might also be the case for lung neuroendocrine neoplasms. However, since multiple molecular subtypes of lung neuroendocrine neoplasms have recently been described, a possible role of polySia-NCAM should be investigated in a larger cohort upon molecular stratification. It may be interesting to examine whether polySia-NCAM expression is more common in certain subtypes of lung neuroendocrine neoplasms.

## 2. Introduction

Polysialic acids (abbr. polySia) are homopolymers of individual sialic acid building blocks. Among them, the polysialylation of the neuronal cell adhesion molecule (abbr. NCAM) is characterized in most detail. Polysialylation of this molecule plays an important role in neurogenesis, and poySia-NCAM remains expressed in plastic brain regions in adults. In addition, it has been shown that polySia-NCAM is expressed on lung neuroendocrine tumors [1], anaplastic Wilms tumor [2], pancreatic tumors [3], aggressive colorectal carcinomas [4], alveolar rhabdomyosarcoma [5] and undifferentiated neuroblastoma [6]. This work focused on the neuroendocrine tumors of the lung, the second largest subgroup of neuroendocrine neoplasms (abbr. NEN) with an incidence of about 25%. They account for approximately 20% of all primary lung tumors and comprise four histological subtypes: typical carcinoids (abbr. TC), atypical carcinoids (abbr. AC), small cell lung carcinoma (abbr. SCLC) and large cell neuroendocrine carcinomas of the lung (abbr. LCNEC). The five year survival rate for SCLC and LCNEC is poor (10 and 15–25%, respectively) [7,8,9,10], whereas that of carcinoids appears slightly better (32%) [11,12]. However, treatment options for metastatic neuroendocrine tumors of the lung are generally very limited.

It has been shown that a high polySia-NCAM expression on neuroendocrine lung carcinomas correlates with early metastasis, presumably by attenuating cell-cell and cell-extracellular matrix contacts [13,14]. Given that, it is has been postulated that there is correlation between PolySia-NCAM positivity and poor prognosis in tumors [13]. In line with this, a positive correlation between polySia-NCAM expression and lymph node metastasis has already been demonstrated in all subtypes of neuroendocrine lung tumors [1]. However, it has so far not been investigated if polySia-NCAM expression correlates with response to therapy and patient prognosis in lung neuroendocrine neoplasms.

## 3. Methods

This work is based on the analysis of digitized patient data and patient samples from the Bundeswehrkrankenhaus Ulm. The main inclusion criterion was the presence of a histologically confirmed neuroendocrine tumor of the lung. Included patients received their diagnosis between 2008 and 2019. Data from 28 patients were analyzed. The approval of the ethics committee was given on 3 May 2021 (ref. no. 147/21), and the research was conducted in accordance with the Declaration of Helsinki.

### 3.1. Imaging

For each patient, a baseline computed tomography image was available at the time of diagnosis. Follow-up CT scans were then obtained at regular, fixed intervals. The RECIST 1.1 criteria were used to assess disease progression [15]. All existing lesions of the tumor (primarius, lymph nodes, metastases and other suspicious masses) were recorded and evaluated. The two largest and most informative lesions of the lung were scored as target lesions, and all other lesions were categorized as non-target lesions. Reference values for partial remission (abbr. PR) were the baseline CT values. For progressive disease (abbr. PD) or stable disease (abbr. SD), the lowest value of the sum diameters within a patient was used as the new reference value. At the end, each patient in the study received an outcome report in terms of a final overall assessment.

### 3.2. Immunohistochemistry

In each case, diagnostic tissue samples were collected by transbronchial forceps biopsy or endobronchial ultrasound-assisted transbronchial needle aspiration (abbr. EBUS-TBNA) and evaluated by conventional light microscopy. Monoclonal antibody 12E3 (mouse IgM) (Thermo Fisher Scientific Inc., Waltham, MA, USA), which binds to polySia of highly polysialylated NCAM, was used as the primary antibodies. Histologic slides of the appendix were on-slide controls. The detection of the monoclonal antibody 12E3 bound to polySia-NCAM was performed using an OptiView DAB IHC Detection Kit with a BenchMark Ultra Autostainer (both from Roche Diagnostics GmbH, Mannheim, Germany). A three-level evaluation score was applied for the assessment of immunostaining: 0 = no polySia, + = weakly positive, ++ = strongly positive. The markers synaptophysin, chromogranin A and CD56 (NCAM) as well as the Ki-67 proliferation index were also recorded.

### 3.3. Statistical Methods

The patient data were initially anonymized and encrypted. Further analyses were performed using GraphPad Prism 9.1.0 (GraphPad Software, San Diego, CA, USA). An χ2 test was used to detect statistical associations between multiple variables. The Kaplan-Meier procedure was used to analyze survival data. A log-rank test was performed to compare the survival probabilities of two groups. The significance level was defined as *p* = 0.05 in each case.

## 4. Results

### 4.1. Baseline Clinical Characteristics of the Study Cohort

Twenty-eight patients were evaluated for polySia-NCAM expression. Most patients presented with SCLC (*n* = 19), and LCNECs were the second most common entity (*n* = 5). Within the group of carcinoids, one patient had a TC, and three patients had an AC. Six of the patients (21.4%) showed a weak or no polySia-NCAM expression. The remaining 22 patients (78.6%) showed a strong staining signal. Figure 1 compares the polySia-NCAM expression within the respective entities of neuroendocrine lung tumors. Within the group of SCLCs, a majority of 84.2% showed a strong polySia-NCAM expression. In the group of LCNECs, there was no polySia-NCAM negative patient. In the carcinoid subtype, the majority of patients (75%) showed a weak or no staining signal for polySia-NCAM. There was a significant correlation between polySia-NCAM expression and high-grade histology (SCLC/LCNEC) (*p* = 0.0140). Some of the tumors were detected at early stages and could be treated curatively. However, the majority of patients received palliative therapy, which varied according to disease progression and type of metastasis (Figure 2). Each of the patient characteristics was tested for a significant correlation with the determined polySia-NCAM expression; however, there was no significant correlation (age: *p* = 0.3405, gender: *p* = 0.6730, smoking history: *p* = 0.1145, ECOC: *p* = 0.1756, UICC stage: *p* = 0.1182), as shown in Figure 3 and Figure 4.

### 4.2. Correlation of Treatment Response with PolySia-NCAM Expression

Treatment response was evaluated according to RECIST 1.1 criteria. In the partial remission (abbr. PR) and progressive disease (abbr. PD) categories, all patients were strongly polySia-NCAM-positive. In the categories of complete remission (abbr. CR) and stable disease (abbr. SD), all patients showed an absent or weak polySia -NCAM expression (Figure 5). Given this, there was no significant correlation between disease progression and polySia-NCAM expression (*p* = 0.0759). Using sphere volumes, the correlation between treatment response and polySia-NCAM expression narrowly failed to reach significance (*p* = 0.0580).

### 4.3. Survival Analysis in Relation to PolySia-NCAM Expression

Analysis of progression-free survival and overall survival was performed in all 28 patients. Six of the patients (21.4%) showed a weak or no polySia-NCAM expression. The patient group with a strong expression of polySia-NCAM included 22 patients (78.6%). Of these, 17 patients died within the observation period of the study. Five of the patients survived beyond the observation period. These five patients all had an approximate observation period of 1000 days. The analysis of progression-free survival was performed for patients with either a progressive disease (abbr. PD) or death depending on which event occurred first. Here, with a *p*-value of 0.4198, no significant correlation could be shown between the two curves with strong and weak or no polySia-NCAM expression (Figure 6). The median survival time in the strongly polySia-NCAM-positive group to the onset of exitus letalis was 411 days. The patient group with no or weak polySia-NCAM expression included six patients (three patients with carcinoid and three with SCLC). Four of these patients died from the disease. The other two patients survived until the end of the study. The median survival time was 578.5 days. There was no significant correlation between polySia-NCAM expression survival (*p* = 0.6918), as shown in Figure 7.

## 5. Discussion

The expression of polySia-NCAM in the different subtypes of lung neuroendocrine tumors has been previously described in the literature. In a 1998 study, Lantouejoul et al. investigated the expression of polySia-NCAM on neuroendocrine lung tumors. In contrast to previous studies, this study also included LCNEC. The authors demonstrated that neuroendocrine tumors differed from other lung tumors, such as adenocarcinoma or squamous cell carcinoma, by the expression of NCAM [1]. All the 28 patients in the present study were also polySia-NCAM positive (cf. Figure 3 and Figure 4). In contrast, two squamous cell carcinomas that were stained as control slides for the present study did not show positivity for NCAM. Lantejoul et al. evaluated polySia-NCAM expression using the monoclonal antibody Mab 735. They used a fine-grained score of 0–300 for evaluation. Their evaluation revealed a significantly higher expression of polySia-NCAM in high-grade tumors in SCLC and LCNEC compared with carcinoids [1]. Similar observations were found in other studies [16]. The results of this work fit well into this context, as a significant correlation between polySia-NCAM expression and high-grade histology was also seen here, although only four well- or moderately differentiated lung endocrine neoplasms were included. Frequently, polySia-NCAM expression on tumors is associated with increased metastasis and late-stage disease [17,18,19,20]. NCAM, as a cell adhesion molecule, can modulate cell-cell and cell-matrix contacts. Expression of polySia on NCAM decreases these adhesion properties. Therefore, it is thought that expression of polySia-NCAM on the surface of tumor cells may promote detachment of such cells and thus promote metastatic spread [19]. In a nude mouse model in which mice were injected with TE671 cells, cells from a cell line of human rhabdomyosarcoma that overexpresses polySia-NCAM, increased formation of metastases was observed. The authors concluded that there was a correlation between polySia-NCAM expression and the metastatic process [21]. Other studies similarly supported the hypothesis that polySia-NCAM expression is related to the metastatic process [13,14]. A more recent study demonstrated that increasing ST8SiaII expression in the SCLC cell line H446 in vitro resulted in increased invasiveness and migration, whereas decreasing ST8SiaII expression had the opposite effect [22]. In keeping with this fact, polySia-NCAM expression is often associated with high-stage disease in the literature. In the present study, however, no such association was found. So far, there is no literature regarding a possible relationship between polySia-NCAM expression and other clinic-pathological characteristics. To verify this, we correlated polySia-NCAM expression with gender, age, ECOG-performance status, and smoking history. Among these criteria, evaluation with respect to gender was of special interest since a recent study showed that leukocytes of men are significantly more polysialylated than those of women [23]. However, as with the other characteristics, no significant correlation was found with respect to gender. So far, it is unclear whether the expression of polySia-NCAM has an impact on patient prognosis, especially within SCLC and LCNEC groups, and whether there is a correlation with treatment response and disease progression. To assess a possible prognostic relevance of polySia-NCAM, a Kaplan-Meier survival analysis was performed, and the extent of the disease was radiologically measured at the time of diagnosis and at certain intervals during therapy. Treatment response was subsequently evaluated according to standard RECIST1.1 and volumetric criteria. However, both the evaluation according to RECIST1.1 and volumetric criteria showed no significant correlation with the expression of polySia-NCAM (RECIST 1.1: *p* = 0.0759; sphere: *p* = 0.0580). Similarly, the difference between the curves for progression-free survival in relation to polySia-NCAM expression (cf. Figure 6, *p* = 0.4198) as well as the evaluation of overall survival in relation to polySia-NCAM expression (cf. Figure 7, *p* = 0.6918) was not significant. As expected, the survival probabilities with respect to the different histological grades were significantly different (*p* = 0.0154). Taken together, while we could confirm the strong expression of polySia-NCAM on high-grade neuroendocrine neoplasms of the lung, we could not find evidence for a possible use of polySia-NCAM immunohistochemistry as a predictive or prognostic biomarker regarding a response to chemotherapy in these tumors. However, the negative results do not change the fact that polySia-NCAM is an interesting target for targeted cancer therapy since it is almost ubiquitously expressed in lung neuroendocrine neoplasms. It has previously been described that the expression of polySia-NCAM is a sign of a poor prognosis [19,24]. Miyahara et al. investigated this association in a study of 17 SCLC patients. In this study, 15 of the 17 patients were NCAM positive and of these 15 patients, another nine patients were positive for polySia-NCAM. Survival probability analysis showed no statistical significance for a better survival prognosis for polySia-NCAM-negative patients, with a *p*-value of 0.0500, but some trend for this hypothesis [25]. In contrast, in the present work, a significantly higher proportion of the 28 neuroendocrine patients evaluated were polySia-NCAM positive (78.6% compared to 52.9% in Miyahara et al.) and the survival curves were also not significantly different with a *p*-value of 0.6918. However, in contrast to Miyahara et al., this analysis was for the entire spectrum of neuroendocrine lung tumors. The study by Lantuejoul et al. also examined the associations between polySia-NCAM expression and survival. This also found that there was no polySia-NCAM score that had an impact on survival [1]. Considering these results, it could be assumed that there is no direct relationship between polySia-NCAM expression and prognosis in neuroendocrine neoplasms of the lung. In summary, polySia expression on NCAM is more common in high-grade NEN of the lung than in low-grade carcinoids. Since polySia-NCAM may still have influence on the invasive properties of lung neuroendocrine tumors, which are not reflected in prognosis, disease progression or treatment response, it might be worthwhile to further investigate the role of polySia-NCAM in early stages of high-grade neuroendocrine neoplasms of the lung.

## 6. Conclusions

In summary, polySia expression on NCAM is more common in high-grade neuroendocrine neoplasms of the lung than in low-grade carcinoids. Although it has undisputed influences on the invasive properties of lung neuroendocrine tumors, which are not reflected in prognosis, disease progression, or treatment response. It is therefore reasonable to speculate that there must be other factors contributing to the invasive properties of the tumors. It may be worthwhile to further investigate the expression of polySia in the context of the molecular level of tumor disease.

## Figures and Tables

**Figure 1 cancers-14-04376-f001:**
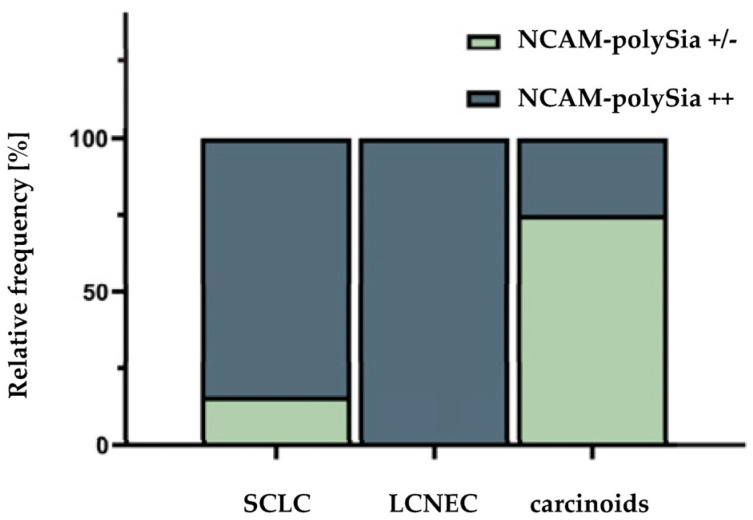
Relationship between the polySia-NCAM expression and entity. Large cell neuroendocrine carcinomas of the lung: LCNEC, small cell lung cancers: SCLC. Within the group of SCLCs, a majority (84.2%) showed strong polySia-NCAM expression. In the group of LCNECs, all patients (100%) were strongly polySia-NCAM positive. In the carcinoid subtype, only 25% of patients showed strong polySia-NCAM expression. An χ2-test was performed to analyze a possible correlation between histology and polySia-NCAM expression (*p* = 0.0140).

**Figure 2 cancers-14-04376-f002:**
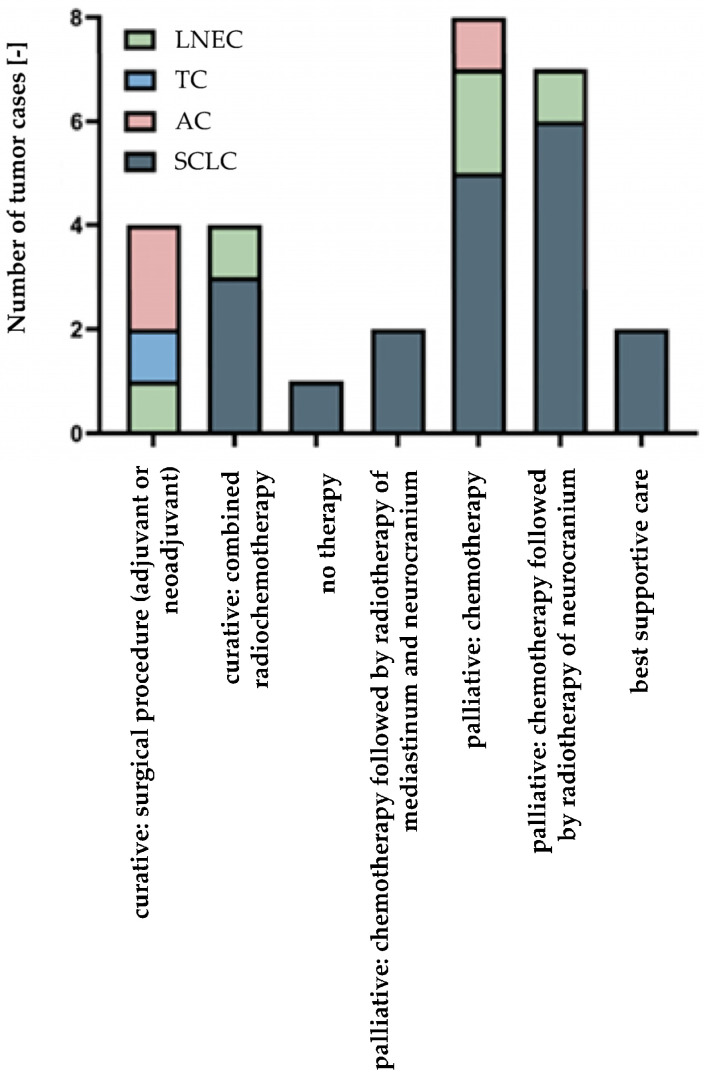
Therapy according to tumor entity. Large cell neuroendocrine carcinomas of the lung: LCNEC, typical carcinoids: TC, atypical carcinoids: AC, small cell lung cancers: SCLC. Three out of four carcinoids were detected relatively early (UICC stage II) and could curatively be treated by an adjuvant or neoadjuvant surgical approach. The fourth patient with a carcinoid received palliative chemotherapy. Two of the five patients with LCNEC could also be curatively treated, and the other three patients received palliative therapy. Within the group of patients with SCLC, three patients showed a limited disease stage at initial diagnosis with curative treatment. All other patients in this group received palliative therapy, which differed according to disease stage.

**Figure 3 cancers-14-04376-f003:**
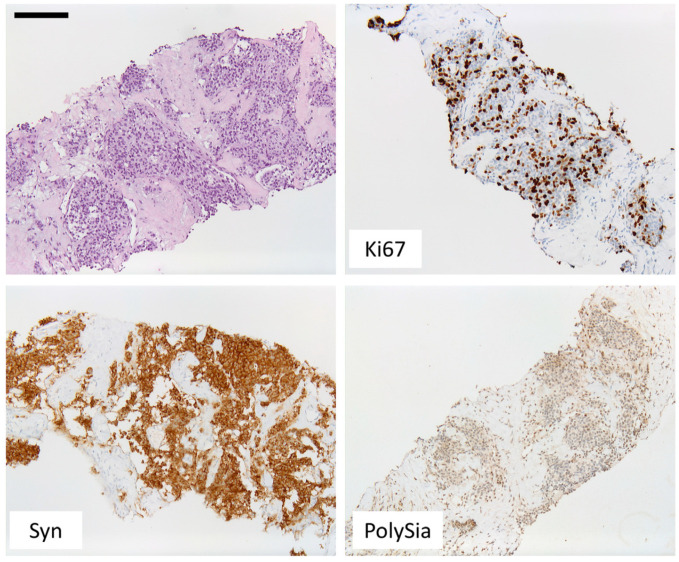
Representative immunohistochemistry of atypical carcinoid (AC) with intermediate proliferation index (Ki-67, 60%), positivity for synaptophysin (Syn) and weak expression of polySia-NCAM (PSS). Bar, 500 µm.

**Figure 4 cancers-14-04376-f004:**
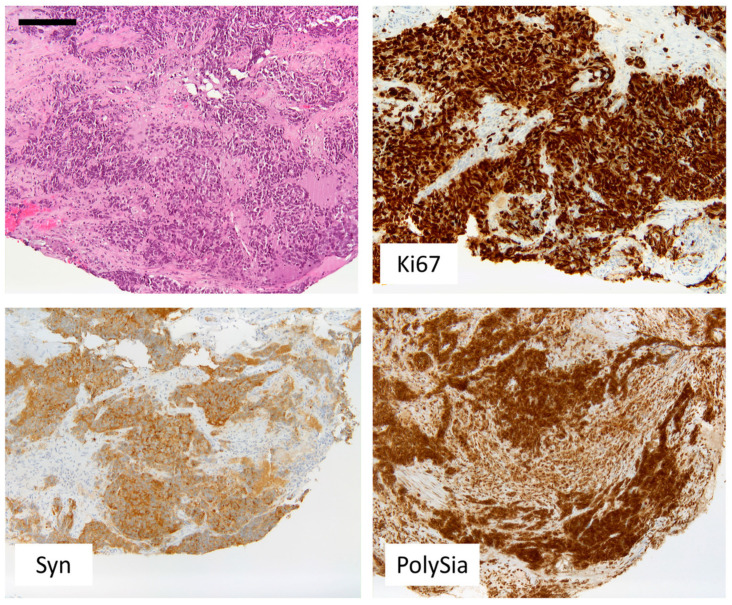
Representative immunohistochemistry of small cell carcinoma (SCLC) with high proliferation index (Ki-67, >95%), positivity for synaptophysin (Syn) and strong expression of polySia-NCAM (PSS). Bar, 500 µm.

**Figure 5 cancers-14-04376-f005:**
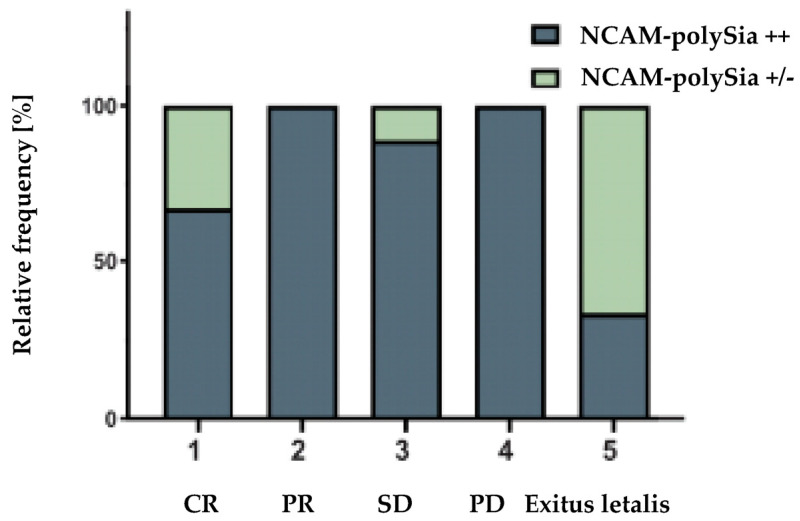
Evaluation of disease progression according to RECIST 1.1 criteria in comparison with the number of patients showing strong polySia-NCAM expression. Most patients had strong expression of polySia-NCAM (grey). While all patients were strongly positive in the category’s partial remission (abbr. PR) and progressive disease (abbr. PD), each patient showed no or only weak polySia-NCAM expression (green) in the category’s complete remission (abbr. CR) and stable disease (SD). In the three patients who died from the disease, only one patient was strongly polySia-NCAM positive.

**Figure 6 cancers-14-04376-f006:**
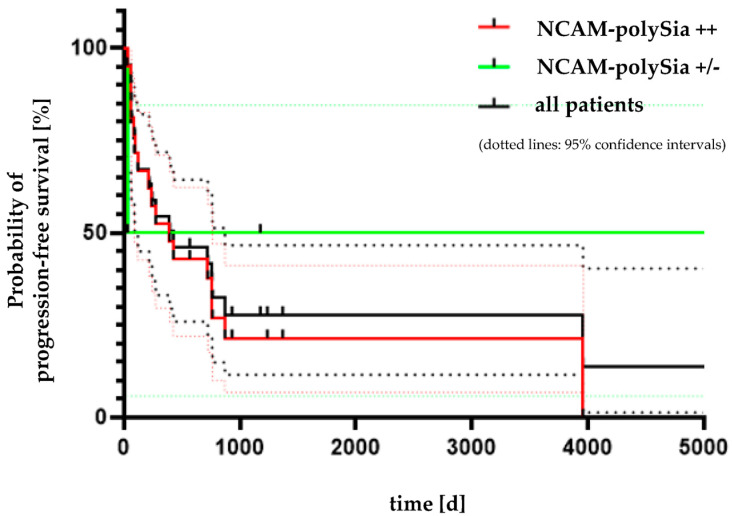
Kaplan-Meier progression-free survival analysis. The black curve shows the progression-free survival of the complete cohort. In the patient group with strong polySia-NCAM expression (red curve), progressive disease (abbr. PD) occurred in five patients. Another 12 patients died from disease. The median time to progressive disease (abbr. PD) or death was 395 days. The patient group with weak or no polySia-NCAM expression (green curve) was underrepresented with four patients. The median survival time was 21,164.5 days. This high number is because the two patients could be followed up for a very long time. With a *p*-value of 0.4198, the log-rank test revealed no significant difference between the two curves according to polySia-NCAM expression.

**Figure 7 cancers-14-04376-f007:**
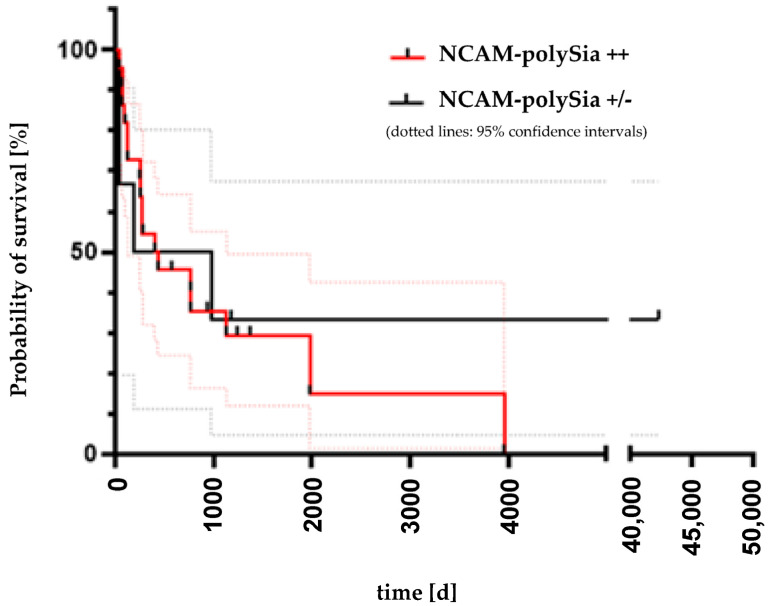
Kaplan-Meier overall survival analysis in relation to polySia-NCAM expression. The patient group with strong polySia-NCAM expression (red curve) included 22 patients. Of these, 17 patients died within the observation period of the study. The median survival time in the strongly polySia-NCAM positive group was 411 days. The patient group with no or weak polySia-NCAM expression (black curve) included 6 patients, 4 of these patients died from disease. The median survival time was 578.5 days. The log-rank test showed no significant difference between the two curves according to polySia-NCAM expression. (*p* = 0.6918).

## Data Availability

No individual participant data will be automatically available. No other documents will be available. Upon request, the underlying dataset will be provided for individual participant data meta-analysis.

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
