# Peer review of "No Impact of PolySia-NCAM Expression on Treatment Response in Neuroendocrine Neoplasms of the Lung"

_cancers, 2022, doi:10.3390/cancers14184376_

Round 1

Reviewer 1 Report

The present manuscript entitled “No impact of polySia-NCAM expression on treatment response in neuroendocrine neoplasms of the lung” by Gagiannis, Gagiannis, Hackenbroch, Horstkotte and Steinestel deals with  a possible correlation between polysialylated NCAM and clinical parameters from patients suffering from neuroendocrine lung tumours. The authors found no major correlation between the presence and/or intensity of polySia signals and treatment response or patient survival. The manuscript is very comprehensive and in parts the descriptions are too short to provide a full understanding for readers.

In the present form, I can not recommend the manuscript for publication.

Comments:

The introduction does not contain a short introduction into neuroendocrine lung tumors.

Many abbreviations used in the manuscript are not explained.

There is no description of sample collection used for immunohistochemistry.

The description of figures 3+4 contains no information about details of samples or stainings. There are no controls shown for comparison.

What kind of endocrine tumour was used? What is PSS?

The length of the bar is never 2.5 micrometer (for comparison a nucleus has between 5 and 10 micrometer in diameter) 

In “Correlation of treatment response with polySia expression” the use of excessive abbreviations does not allow the understanding of the described context. Putting abbreviations in quotation marks does not explain anything. 

The chapter “survival analysis in relation to polySia-expression” (by the way a typological error using the hyphen) is not logical:

The patient group with 22 individuals seems to be the LCNEC (line 145). Splitting these patients in “strong and weak/no polySia” contradicts the results shown in Figure 1, where all patients from the LCNEC group were found polySia positive. What is the explanation?

Author Response

POINT-BY-POINT RESPONSE

Comments to the Author

The present manuscript entitled “No impact of polySia-NCAM expression on treatment response in neuroendocrine neoplasms of the lung” by Gagiannis, Gagiannis, Hackenbroch, Horstkotte and Steinestel deals with a possible correlation between polysialylated NCAM and clinical parameters from patients suffering from neuroendocrine lung tumours. The authors found no major correlation between the presence and/or intensity of polySia signals and treatment response or patient survival. The manuscript is very comprehensive and in parts the descriptions are too short to provide a full understanding for readers. In the present form, I can not recommend the manuscript for publication.

  1. The introduction does not contain a short introduction into neuroendocrine lung tumors.

We thank you for this constructive advice. A short introduction on the topic of neuroendocrine neoplasms of the lung has been added to the introduction section of the revised manuscript.

  1. Many abbreviations used in the manuscript are not explained.

We have thoroughly reviewed the manuscript and explained all abbreviations in the context of their respective use.

  1. There is no description of sample collection used for immunohistochemistry.

We apologize for this fact and have included this information in the ‘Materials and Methods’ section of the revised manuscript.

  1. The description of figures 3+4 contains no information about details of samples or stainings. There are no controls shown for comparison. What kind of endocrine tumour was used? What is PSS? The length of the bar is never 2.5 micrometer (for comparison a nucleus has between 5 and 10 micrometer in diameter) 

Thank you for your comment and for pointing out the mistake in the previous version of the figure legend concerning the length of the bar in the microphotographs. The Figure legends for figures 3 and 4 have been revised and, in the current version of the manuscript, include all relevant information on tumor entity (atypical carcinoid and small cell lung carcioma, respectively), stainings (Ki67, Synaptophysin and polySia-NCAM) and length of the scale bar (500µm). For all stainings, tissue sections from human appendix (Ki67, Synaptophysin) and tonsil (polySia-NCAM) were used as positive and negative on-slide controls.

  1. In “Correlation of treatment response with polySia expression” the use of excessive abbreviations does not allow the understanding of the described context. Putting abbreviations in quotation marks does not explain anything. 

Analogous to point 2, we would like to expressly apologize once again for the frequent use of abbreviations. We have corrected this problem and hope that this has improved the flow of reading.

  1. The chapter “survival analysis in relation to polySia-expression” (by the way a typological error using the hyphen) is not logical: The patient group with 22 individuals seems to be the LCNEC (line 145). Splitting these patients in “strong and weak/no polySia” contradicts the results shown in Figure 1, where all patients from the LCNEC group were found polySia positive. What is the explanation?

We are sorry that this section was not clearly worded. Figure 1 shows that all large cell neuroendocrine carcinomas of the lung (LCNEC) showed strong polySia-NCAM expression. However, LCNEC comprised only 5 of 28 patients (total population) Besides LCNEC, the 22 strongly positive samples included SCLC. mainly SCLC patients. We changed both the description of Figure 1 and the respective paragraph, which should improve readability and internal stringency of the text.

Reviewer 2 Report

In the current study titled, “No impact of polySia-NCAM expression on treatment response in neuroendocrine neoplasms of the lung” authors tried to investigate if there is any correlation between polySia-NCAM expression and clinic-pathological characteristics of lung neuroendocrine neoplasms. But, against the assumption, the authors did not find any correlation between the polySia-NCAM expression and the prognosis of the disease. I find the study very intriguing and agree with the authors that there must be a thorough investigation of this issue with a bigger cohort. Overall, I agree with the study design and interpretation of the observations. However, I believe, authors need to explain these findings in more detail and with more description. I believe some of the results especially mentioned in figure 3 and figure 4 must be discussed in more detail. I have the following comments and edits.

Comment

Comment 1: Please clearly define all the terms where abbreviations are used

Comment 2: Please explain figure legends properly. Add a few sentences explaining each figure/result. Please fix the fonts for all figures. Should be consistent with the text.

Comment 3: Figure 2: Please clearly define the Y axis. Figure 6 and Figure 7: Please define all the plot lines

Comment 4: If Figure 3 and Figure 4 have a representative tissue sample each then the individual sections must be oriented properly. If not then can authors share the images from the same tissue for each individual marker.

Comment 5: Please quote corresponding figures from the article while discussing the results in the discussion section

Author Response

POINT-BY-POINT RESPONSE

Comments to the Author

In the current study titled, “No impact of polySia-NCAM expression on treatment response in neuroendocrine neoplasms of the lung” authors tried to investigate if there is any correlation between polySia-NCAM expression and clinic-pathological characteristics of lung neuroendocrine neoplasms. But, against the assumption, the authors did not find any correlation between the polySia-NCAM expression and the prognosis of the disease. I find the study very intriguing and agree with the authors that there must be a thorough investigation of this issue with a bigger cohort. Overall, I agree with the study design and interpretation of the observations. However, I believe, authors need to explain these findings in more detail and with more description. I believe some of the results especially mentioned in figure 3 and figure 4 must be discussed in more detail. I have the following comments and edits.

  1. Please clearly define all the terms where abbreviations are used.

We have thoroughly reviewed the manuscript and explained all abbreviations in the context of their respective use (see response 1 to reviewer #1).

  1. Please explain figure legends properly. Add a few sentences explaining each figure/result. Please fix the fonts for all figures. Should be consistent with the text.

We would also like to take this opportunity to thank you for your constructive suggestion for improvement. We have corrected the figure labels as well as the fonts and hope to have significantly improved the quality of the content.

  1. Please clearly define the Y axis. Figure 6 and Figure 7: Please define all the plot lines.

Analogous to point 2, we have improved illustrations accordingly.

  1. If Figure 3 and Figure 4 have a representative tissue sample each then the individual sections must be oriented properly. If not then can authors share the images from the same tissue for each individual marker.

Thank you for this valuable comment. Figures 3 and 4 are microphotographs from consecutive stainings (HE, Ki67, Synaptophysin, polySia-NCAM) from identical tumor samples (atypical carcinoid and small cell carcinoma, respectively). While all microphotographs in Figure 4 are now oriented properly, since the orientation of the core needle biopsy containing AC was different on all slides, this was not possible for Figure 3. However, the figure legends for Figures 3 and 4 have now been revised to improve clarity.

  1. Please quote corresponding figures from the article while discussing the results in the discussion section.

Thank you. Referencing the figures in the discussion section makes the text much more understandable and improves readabilty. This has been revised accordingly in the current version of the manuscript.
